# Computational Study of Tesla Valve Design for Vesico-Amniotic Shunt to Manage Lower Urinary Tract Obstruction and Pleural Effusion

**DOI:** 10.3390/bioengineering12101126

**Published:** 2025-10-21

**Authors:** SaiSri Nakirekanti, Varun Chandra Sarkonda, Janet Dong, Donglu Shi, Ahmad M. Alsaghir, Je-Hyeong Bahk, Braxton Forde

**Affiliations:** 1Department of Mechanical and Materials Engineering, College of Engineering and Applied Science, University of Cincinnati, Cincinnati, OH 45221, USA; nakiresi@mail.uc.edu (S.N.); sarkonva@mail.uc.edu (V.C.S.); alsagham@mail.uc.edu (A.M.A.); bahkjg@ucmail.uc.edu (J.-H.B.); 2Department of Biomedical Engineering, College of Engineering and Applied Science, University of Cincinnati, Cincinnati, OH 45221, USA; shid@ucmail.uc.edu; 3Department of Obstetrics and Gynecology, College of Medicine, University of Cincinnati, Cincinnati, OH 45267, USA; fordebn@ucmail.uc.edu

**Keywords:** LUTO, fetal shunt, vesico-amniotic shunt, Tesla valve, CFD, diodicity, Reynolds number

## Abstract

Fetal lower urinary tract obstruction (LUTO) and pleural effusion are conditions that can disrupt fetal growth and lead to fetal death. LUTO inhibits the formation of amniotic fluid, which is vital for lung development, while pleural effusions can compress the fetal heart, potentially causing fatal cardiac failure. To manage these conditions, a fetal shunt (vesico-amniotic shunt) is placed inside the fetal bladder. This paper presents a study on a new design incorporating a Tesla valve in the shunt. Six groups of Tesla valves with loop angles of 50 degrees and 60 degrees, and different end dimensions, are examined and evaluated in terms of the urine flow rate from the fetal bladder into the amniotic cavity, the pressure buildup between the two sides, and their potential in developing fetal bladder muscles. A mathematical method is used to compare diode characteristics, analyze flow rates, identify the Tesla valve angle, determine the Reynolds number, and assess diodicity. The Computational Fluid Dynamics (CFD) method is also employed to verify calculation results and simulate fluid behavior inside the Tesla valve. Combining the calculations and simulations, a 50-degree Tesla valve with specific dimensions showed the best performance and will be the optimal design for the fetal shunt.

## 1. Introduction

Fetal lower urinary tract obstruction (LUTO) and fetal pleural effusion complicate up to 1 in 5000 and 1 in 10,000 pregnancies, respectively [1]. In the setting of LUTO, the amniotic fluid, which the fetus floats in, primarily consists of fetal urine and is required for proper fetal lung development. With an obstruction in the lower urinary tract, there is no production of amniotic fluid, leading to pulmonary hypoplasia and resultant neonatal death [2]. The causes of LUTO are highly dependent on fetal gender. In males, the most common cause is posterior urethral valves (PUVs), while in females, it is typically urethral atresia [1]. In the setting of pleural effusions, the fluid is trapped inside the fetal thorax, which leads to compression of the fetal heart and lungs, resultant fetal cardiac failure, and intrauterine fetal death [3]. The condition of pleural effusions may be primary due to lymphatic maldevelopment or secondary to other issues, such as genetic syndromes, infections, or malformations, leading to non-immune hydrops fetalis [3].

All above conditions have been treated with in utero fetal therapy via vesico-amniotic shunt placement [4,5,6]. The in utero fetal shunting redirects the fluid to the amniotic cavity, allowing for normalization of the flow of amniotic fluid in the setting of LUTO or decompression of the thorax. The currently commercially available shunts, however, have significant limitations, such as the high rates of shunt dislodgement as well as the inability to create a flow gradient between the area being shunted and the amniotic cavity [7]. In the setting of vesico-amniotic shunting or thoraco-amniotic shunting, this lack of pressure gradient can significantly impact both pre- and post-natal outcomes [8]. For thoraco-amniotic shunts, this lack of a pressure gradient may lead to shunt failure and persistence of pleural effusion, preventing proper improvement in clinical status [9]. For LUTO, the issue is more pronounced, as the equilibration of pressure between the amniotic sac and the bladder leads to a high resting tone in the fetal bladder, and subsequently scarring and further bladder dysfunction [9]. Additionally, shunting has failed in previous disease states such as severe fetal ventriculomegaly secondary to an inability to create a driving pressure out of the head, as well as the risk of meconium flowing back into the fetal ventricles [10,11].

The currently available fetal shunt in the United States is the Harrison fetal bladder stent [12,13] (Figure 1a). Once it is deployed, the shunt simply connects an internal fetal compartment to the amniotic space, so there is an equilibration of pressure between the two compartments. Additionally, the fluid that is located in the amniotic space can also make its way into the fetal space. This constant fluid exchange prevents normal bladder training for the bladder to learn coordinated contractions. A one-way valve would better simulate the fetal urethra and would allow for proper fetal bladder development and decrease fetal bladder scarring, which can be a lifelong morbidity [14,15].

Outside the United States, the available shunts are all limited by this lack of pressure gradient creation. This has led to the ongoing development of the vortex shunt (Figure 1b), which aims to address the dislodgement risk associated with current shunts. Instead of creating a pressure gradient, it features a one-way valve [16,17]. This may not correct the pressure issues for a few reasons. Firstly, this shunt has been developed for normal bladder priming and is based upon the filling and empty pressures of healthy bladders [16,17]. However, given the stretching and dysfunction secondary to LUTO, the pressures in the bladder are significantly higher. Additionally, these bladder pressures are different than those of the fetal brain and the fetal thorax [18], with the fetal thorax’s resting pressure being far below that of normal amniotic fluid (28.549 cm-H_2_O) and much higher than that when a large effusion is present (53 cm-H_2_O). Thus, there is a significant unmet need for the development of a fetal shunt that creates a gentle pressure gradient across various pressure settings to drive fluid from inside the fetus to the amniotic space.

The purpose of this paper is to address the critical limitations of current fetal shunt technologies, particularly their inability to generate a sufficient and controlled pressure gradient between fetal compartments and the amniotic cavity—an issue that compromises outcomes in conditions such as LUTO, fetal pleural effusions, and ventriculomegaly. Current shunts, including the Harrison fetal bladder stent and recently developed vortex shunts, either suffer from high dislodgement rates or fail to accommodate the varied pressure environments of different fetal organs. To overcome these challenges, this study investigates the use of a Tesla valve—a passive, no-moving-parts flow regulator—as a potential design element for a next-generation fetal shunt. Through Computational Fluid Dynamics (CFD) modeling, the study evaluates how variations in Tesla valve geometry and shunt length influence pressure gradients under physiologically relevant conditions. The ultimate goal is to design a fetal shunt that can provide reliable, directionally biased fluid flow tailored to the diverse pressure requirements of fetal therapy.

## 2. Materials and Methods

### 2.1. New Vescio-Amniotic Shunt Design

To overcome current shunt limitations, we propose designing a new fetal shunt (acting as a vesico-amniotic shunt) with a Tesla valve inside—named the Double Mushroom Tesla Fetal Shunt. The proposed shunt design is illustrated in Figure 2. There are several sections of the shunt. Sections E + F are placed into the fetal bladder, facilitating efficient urine drainage. Sections H + I are the terminal sections of the shunt that extend into the amniotic cavity. Sections E + F and H + I are designed like mushrooms. Mushroom design helps minimize the risk of dislodgement, reducing the need for re-interventions, and lowering the risk of premature rupture of membranes [19]. The ends of the shunt are developed for particular purposes based on the anatomical placement. The middle portion of the shunt, designated as section G, consists of two layers. The outside layer, along with sections E, F, H, and I, is a mesh that is fabricated from braided nitinol wire. This braided structure encloses and facilitates the inner layer—Tesla valve body (T)—which contains the passage of fluid through the internal lumen, as shown in Figure 2a,c.

A Tesla valve is a passive, no-moving-parts fluidic device that allows fluid to flow more easily in one direction than the other by utilizing a series of asymmetric flow channels [20,21]. There are two directions of fluid flow: forward (indicated in blue color) and reverse or backward (indicated in red color), as shown in Figure 2b. In our new shunt design, the forward flow is defined as the flow from the fetal bladder to the amniotic cavity, and the backward flow is defined as the flow from the amniotic cavity to the fetal bladder. By adjusting the geometry of a Tesla valve, the pressure gradient is subsequently altered. Tesla valves offer a significant opportunity for creating the optimal pressure gradients required for various utilities in fetal surgery. We thus sought to evaluate, via mathematical calculations and CFD method, the impact of shunt length and Tesla valve angle on pressure gradients, considering known pressures in fetal compartments to design the optimal Tesla valve for use in fetal therapy.

### 2.2. Tesla Valve Geometry

The geometric parameters of a single loop of Tesla valve are shown in Figure 3a, including the loop angle *θ*, segment lengths *L*_1_ and *L*_2_, and arc radius *R*. The full length of the Tesla valve in the new shunt is shown in Figure 3b. Section *A* is at the Fetal bladder side (inlet) and section *B* is at the Amniotic cavity side (outlet). The fluid channel width of the Tesla valve is 0.4 mm, and the diameter of the shunt or Tesla valve body is 2.6 mm. We designed the Tesla valve using the formulated geometric equations [22].R = L_1_ tan θ,(1)L_2_ = R + L_1_/cos θ,(2)
where *L*_1_ and *L*_2_ are the lengths of the segments of Tesla valves and *R* is the arc radius. With these relations, several loop angles *θ* are used to determine the best angle for the fetal shunt application. 70 and 80-degree loop angles cause the size of the loops to exceed the required diameter (2.6 mm) for the shunt. The angles of 50 degrees and 60 degrees were preferred because they create geometric loops that are suitable for the shunt, allowing the effects of angle on diodicity, pressure distribution, velocity variation, and flow rate to be obtained. The Reynolds number varies from 1675 to 2094 using Equation (4). The diodicity (*Di*) of the Tesla valve is used to measure its effectiveness in forward and backward directions. This is the ratio of pressure drop in reverse flow (∆*Pr*) to forward flow (∆*Pf*) [23].*Di* = (∆*Pr*)/(∆*Pf*)(3)

The Reynolds number (Re) is defined as [24],*Re* = (*ρνDh*)/*μ*(4)
where *ν* is the inlet velocity, *D_h_* is the hydraulic diameter, *ρ* is the density, and *μ* is the kinematic viscosity, respectively.

### 2.3. Quantifying Valve Performance with Diodicity

Generally, diodicity is defined as the ratio of the pressure difference in the forward direction to the backward direction at a constant flow rate [22]. However, we redefined it as the ratio of the flow rate in the forward direction (*Qf*) to the backward direction (*Qr*) when the applied pressure difference is the same across the forward and backward directions. In this application, the Tesla valve is used in the shunt to drain the urine from the fetal bladder to the amniotic cavity due to a natural pressure difference. This is the most precise method for calculating the efficiency of the Tesla valve in preventing backward flow and enabling forward flow.*Di* = (*Qf*)/(*Qr*)(5)

#### 2.3.1. Mesh Independence Study

Three-dimensional numerical simulation analysis is used to investigate the performance of the presented Tesla valves. To ensure the numerical accuracy of the simulation, a grid independence study was conducted. The flow domain was discretized using an unstructured triangular mesh (Figure 4b), and the mesh was systematically refined. The study confirmed that simulation results, such as diodicity, became independent of mesh density at approximately 1.26 million elements (or 1.26 × 10^6^ cells) for the 50° valve and 1.1 million elements for the 60° valve, as illustrated in the graph in Figure 4c. As further refinement in the mesh did not yield significantly different results, these mesh densities were adopted for all final simulations.

#### 2.3.2. Numerical Methods with Fluid Properties and Assumptions

The working fluid simulates fetal urine and was modeled as an incompressible Newtonian fluid. The fluid properties were assumed to be equivalent to those of water at body temperature (37 °C), with a density (ρ) of 993 kg/m^3^ and a dynamic viscosity (μ) of 0.000691 Pa·s [25].

#### 2.3.3. Governing Equations and Turbulence Model

The simulations were performed using Ansys Fluent. A turbulent, incompressible, steady-state solver, along with the SIMPLE (Semi-Implicit Method for Pressure Linked Equations) algorithm, was utilized to solve the governing system of equations [26]. The simulation solves the Reynolds-Averaged Navier–Stokes (RANS) equations, which are as follows:(6)$$\nabla \cdot \vec{V}=0$$
(7)$$\rho (\vec{V}\cdot \nabla )\;\vec{V}=-\nabla P+\mu \nabla^{2}\vec{V}-\nabla \rho v^{\prime }v^{\prime }$$
where $\vec{V}$ is the instantaneous velocity vector ($\vec{V}$ =$\;\vec{V}$ + v′), *μ* is the dynamic viscosity of the fluid (mention the fluid used in this case), *P* is the static pressure, and *ρv′v′* is the Reynolds stress term [27]. The standard *k-ε* turbulence model is used to solve for this term. The corresponding governing equations are as follows,(8)$$\rho \left(\vec{V}\cdot \nabla \right)k=\nabla \left[\left(\mu +\frac{\mu_{t}}{\sigma_{k}}\right)\nabla k\right]+G_{k}+G_{b}-\rho \varepsilon -Y_{M}+S_{k}$$(9)$$\rho \left(\vec{V}\cdot \nabla \right)\varepsilon =\nabla \left[\left(\mu +\frac{\mu_{t}}{\sigma_{\varepsilon }}\right)\nabla \varepsilon \right]+C_{1k}\frac{\varepsilon }{k}\left(G_{k}+C_{3\varepsilon }C_{b}\right)-C_{2\varepsilon }\rho \frac{\varepsilon^{2}}{k}+S_{\varepsilon }$$(10)$$\mu_{t}=\rho C_{\mu }\frac{k^{2}}{\varepsilon }$$
where *k* is the turbulent kinetic energy, *ε* is the rate of turbulence dissipation, $\mu_{t}\;$ is turbulent viscosity, $C_{1k}$, $C_{2\varepsilon }$, $C_{3\varepsilon }$, $C_{b}$ are constant, and $\sigma_{k}$ and $\sigma_{\varepsilon }\;$ are Prandtl numbers [28]. Given that the Reynolds number for the flow ranges from 1675 to 2094, the flow is in the transitional regime. The k-ε model was therefore chosen to accurately capture the potential onset of turbulent characteristics.

#### 2.3.4. Boundary Conditions and Convergence Criteria

The boundary conditions were based on clinically relevant neonatal bladder pressures. The inlet pressure on the fetal bladder side was varied from 20 to 40 cm-H_2_O to simulate active voiding. The outlet pressure on the amniotic cavity side was set to 7–13 cm-H_2_O. A no-slip condition was applied to all internal walls of the shunt. The simulation was solved using the SIMPLE algorithm and was considered converged when solution residuals reached a value of 10^−5^ [21].

### 2.4. Six Design Cases for Simulations

The new double mushroom Tesla shunt design with dimensions is illustrated in Figure 5a. The total length is at least 25 mm, including the lengths of the bulbous regions at both ends: the fetal bladder side (5 mm) and the amniotic cavity side (5 mm), along with the Tesla body length of approximately 15 mm, which serves as the base length. The key scientific methods to evaluate the performance of the Tesla valve include pressure, velocity distributions, flow rate calculations, and diodicity under boundary conditions concerning the fetal bladder, with Reynolds numbers ranging from 1675 to 2094. In this paper, we tested two sets of three different models of the Tesla valve with a total of six cases. One set is for a 50-degree loop angle Tesla valve with three varied-length extensions on both ends of the fetal bladder and the amniotic cavity. The other set is for a 60-degree loop angle Tesla valve with three varied-length extensions on both ends. To ensure proper flow immediately from the fetal bladder, the 2 mm, 1 mm, and 0.5 mm extensions are considered, as described in the following cases.

Cases 1 and 2: Asymmetrical extension on both ends of the shunt.

The base length for the model is 15 mm, denoted by L. The original length on the fetal bladder side is assumed as A, and on the amniotic cavity is assumed as B. The new length on the fetal bladder side is A = 1 mm, and the amniotic cavity side is B = 2 mm, as shown in Figure 5b. This asymmetrical extension aims to evaluate how differences in extension lengths influence the valve’s diodicity and flow resistance. Case 1 is for a 50-degree Tesla valve, and Case 2 is for a 60-degree Tesla valve.

Cases 3 and 4: Symmetrical extension of 2 mm on both ends of the shunt.

In this model, it is considered as A = 2 mm on the fetal bladder side and the amniotic cavity side as shown in Figure 5c. This base setup aims to evaluate the diodicity under the boundary conditions of a pressure inlet (20–40 cm-H_2_O) on the fetal bladder and a pressure outlet (7–13 cm-H_2_O) on the amniotic cavity side (amniotic space). Case 3 is for a 50-degree Tesla valve, and Case 4 is for a 60-degree Tesla valve.

Cases 5 and 6: Predominant extension on the amniotic cavity side.

In cases 5 and 6, extension A is 0.5 mm and extension B is 2 mm, as shown in Figure 5d, which shows the predominant extension on the amniotic cavity side. Case 5 is for a 50-degree Tesla valve, and Case 6 is for a 60-degree Tesla valve.

## 3. Results

This section presents the simulation results for the abovementioned case studies, including Tesla loop angle on flow rate, diodicity analysis, pressure contour analysis, velocity contour analysis, and vector analysis. The interpretations of pressure and velocity contour results are also presented.

### 3.1. Geometric Impact on Tesla Valve Flow Rate

The CFD simulation method is used to assess the impact of two sets of cases on diodicity and flow rate. The graphs illustrate the volume flow rate as a function of the applied pressure difference, along with various geometric parameters, as shown in Figure 6. The pressure difference applied across the Tesla valve in the simulations is similar to that found between the fetal bladder and the amniotic cavity. Two different Tesla valve angles, 50 degrees and 60 degrees, were examined. Within these angles, the effect of horizontal extensions on both sides of the Tesla valve is evaluated among the three configurations described in cases 1, 3, and 5 for the 50-degree Tesla valve, and cases 2, 4, and 6 for the 60-degree Tesla valve. Results indicate that these extensions significantly influence the volume flow rate. Among these extensions, Cases 5 and 6, with 0.5 mm on the fetal bladder side and 2 mm on the amniotic cavity side, exhibit the highest volume flow rate compared to other dimensions, as illustrated by the blue lines in Figure 6a–d, regardless of whether it’s a 50-degree or 60-degree valve, and irrespective of the fluid’s travel direction, forward or backward. The demonstration of the highest volume flow rate among the alternative dimensions is due to the minimal flow disturbances and energy losses as fluid passes the side branch junctions in the forward direction, thereby reducing overall flow resistance.

While the Tesla valve’s channel and design inherently reveal its contribution to the flow characteristics, optimizing the horizontal extensions allows us to achieve the maximum flow in the forward direction, as shown by the graphs in Figure 6a,b. In the backward direction, as shown in Figure 6c,d, volume flow rates were also measured for the same conditions and pressure range as that of forward flow, the 0.5 mm fetal bladder side with 2 mm amniotic cavity side geometry also tended to have the highest reverse volume flow rates among the geometries tested, while the 2 mm on both sides geometry tended to have the lowest reverse flow rates. Both the inclination or loop angle of the valve and the horizontal dimensions of the extensions have a very significant function in deciding the quantity of flow in both directions, and that suggests the necessity of maximizing these geometric parameters to control fluid flow. 0.5 mm on the fetal bladder side and 2 mm on the amniotic cavity side are the best dimensions to achieve the required flow rates in both 50- and 60-degree valves.

### 3.2. Diodicity Analysis

Figure 7a,b are the plots of diodicity vs. pressure differences for a 50-degree Tesla valve and a 60-degree Tesla valve, respectively. There are three sets of lines for three difference case setups, with blue line for the geometric design of 0.5 mm on the fetal bladder side and 2 mm on the amniotic cavity side, which exhibit the highest diodicity for any pressure differences, despite loop angles (50 and 60 degrees) of the Tesla valve, among the tested geometries. 2 mm on both sides (fetal bladder and amniotic cavity) shows the lowest diodicity. This indicates that while showing higher resistance to flow in both directions makes it less effective for creating the resistance between the forward and backward flows. From these graphs, it is evident that horizontal extensions and angles have a significant effect on diodicity.

### 3.3. Pressure Contour Analysis: Directional Flow Resistance

Based on the flow rate and diodicity analysis in the previous two sections, Tesla valve designs in cases 5 and 6 demonstrate the best performance regarding flow rate and diodicity. Therefore, this section will concentrate solely on pressure contour analysis for cases 5 and 6.

#### 3.3.1. Pressure Contour Analysis of 50-Degree Tesla Valve (Case 5)

Figure 8a–c illustrate the pressure contours in the forward flow direction, while Figure 8d–f depict the pressure contours in the backward direction, for a 50-degree valve. The pressures applied to the fetal bladder side are 20, 30, and 40 cm-H_2_O, respectively, and the pressures applied to the amniotic cavity side are 7, 12, and 13 cm-H_2_O, respectively, in the forward pressure contour simulations. For the backward pressure contours, the pressures are applied to each end, vice versa. As shown in Figure 8a–c, the color bar represents the magnitude of pressure in pascals, where vibrant colors (reds, oranges, yellows) indicate higher-pressure regions, and serene colors (green, cyan, blue) indicate lower-pressure regions. The flow is from left to right, with a transition slowly from warmer colors to cooler colors, and there is no sudden variation among different colors, which means that the pressure drop is not abrupt but occurs progressively along the entire length of the valve. Therefore, the forward flow of fluid direction from the fetal bladder to the amniotic cavity exhibits a relatively smooth and gradual decrease in static pressure along the length of the valve. Notably, there are no significant regions of high-pressure buildup or abrupt pressure drops within the internal loops of the valve. This suggests that the valve geometry offers comparatively low resistance to flow in the forward direction, allowing for flow rates of 10.518, 12.858, and 16.692 (mL-min) under applied pressure differences of 13, 18, and 27 cm-H_2_O in the forward direction.

Figure 8d–f show a sudden variation between different colors, particularly at the turns, when the flow is reversed. This indicates a significantly high-pressure buildup concentrated within the loops. There is a substantial increase in resistance to flow in a backward direction. This suggests that the valve geometry offers comparatively high resistance to flow in the backward direction, allowing for a flow rate of 9.72, 11.76, and 15.054 (mL-min) under the applied pressure differences of 13, 18, and 27 cm-H_2_O in the backward direction.

#### 3.3.2. Pressure Contour Analysis of 60-Degree Tesla Valve (Case 6)

Figure 9a–c represent the pressure contours in the forward direction, and Figure 9d–f represent the pressure contours in the backward direction for a 60-degree Tesla valve. The same set of boundary conditions used in the 50-degree simulations is applied to the 60-degree Tesla valve simulations. As shown in Figure 9a–c, the color bar represents the magnitude of pressure in pascals, where vibrant colors (reds, oranges, yellows) indicate higher-pressure regions, and serene colors (green, cyan, blue) indicate lower-pressure regions. The flow is from left to right, with a transition slowly from warmer colors to cooler colors, and there is no sudden variation among different colors, which means that the pressure drop is not abrupt but occurs progressively along the entire length of the valve. Therefore, the forward flow of fluid direction from the fetal bladder to the amniotic cavity exhibits a relatively smooth and gradual decrease in static pressure along the length of the valve. Notably, there are no significant regions of high-pressure buildup or abrupt pressure drops within the internal loops of the valve. This suggests that the valve geometry offers comparatively low resistance to flow in the forward direction, allowing for flow rates of 9.87, 11.85, and 15 (mL-min) under applied pressure differences of 13, 18, and 27 cm-H_2_O in the forward direction. Figure 9d–f show a significantly high-pressure buildup concentrated within the loops, particularly at the turns, when the flow is reversed. This indicates a substantial increase in resistance to flow in the backward direction. This suggests that the valve geometry offers comparatively high resistance to flow in the backward direction, allowing for flow rates of 9.26, 10.99, and 13.62 (mL-min) under applied pressure differences of 13.256, 18, and 27 cm-H_2_O in the backward direction. The difference in the values of flow rate is noticeable due to the number of loops and the length of the loop in each Tesla valve. In the case of the 50-degree valve, five loops are present with a larger length segment compared to the length segment of the loop in the 60-degree valve, which has six loops. If the number of loops exceeds six, as in the 60-degree valve, more pressure losses will occur, resulting in increased resistance to flow and a reduction in the overall flow rate.

### 3.4. Velocity Contour Analysis

Based on the flow rate and diodicity analysis in the previous two sections, Tesla valve designs in cases 5 and 6 demonstrate the best performance regarding flow rate and diodicity. Therefore, this section will concentrate solely on velocity contour analysis for cases 5 and 6.

#### 3.4.1. Velocity Contours of 50-Degree Tesla Valve (Case 5)

Figure 10a–c represent the velocity contours in the forward direction for the 50-degree Tesla valve, and the fluid flows from left to right (fetal bladder side to amniotic cavity side). The pressures applied to the fetal bladder side are 20, 30, 40 cm-H_2_O, and the pressures applied to the amniotic cavity side are 7, 12, 13 cm-H_2_O, respectively. As an example, Figure 10a represents the velocity contour of the valve with 20 cm-H_2_O pressure applied to the fetal bladder side and 7 cm-H_2_O pressure applied to the amniotic cavity side. In Figure 10a–c, the red color represents the highest velocity magnitude, the blue color represents the lowest velocity magnitude, and the colors orange, yellow, and green represent the velocity magnitude between the highest and lowest. At the entrance of the valve (fetal bladder side), the initial velocity of the fluid is low, as indicated by the orange and yellow colors in the straight channel. It speeds up as it moves forward and can be visualized by the several straight, red-colored sections, which represent a continuous velocity path. As the loops are designed to restrict the flow, they exhibit very low velocity magnitudes as shown in the blue-colored zones within the loops.

Figure 10d–f depict the velocity contours in the backward direction. The pressures applied to the amniotic cavity side are 20, 30, 40 cm-H_2_O, and the pressures on the fetal bladder side are 7, 12, 13 cm-H_2_O, respectively. As an example, Figure 10d illustrates the velocity contour of the valve with 7 cm-H_2_O pressure applied to the fetal bladder side and 20 cm-H_2_O pressure on the amniotic cavity side. The fluid is simulated to enter from the right side (amniotic cavity) to the left (the fetal bladder). As shown in Figure 10d–f, the red color is present near the entrance on the right side, indicating a high-velocity region. As the flow continues, it does not move in a straight channel; instead, it flows in a broken fashion, as seen in the figure of broken red zones. The red zones are not continuous, nor in a straight fashion, with some red appearing in the curved sections (loops). The flow is also not even steady in the loops, as evidenced by the non-continuous red-orange areas. Therefore, the flow path is not sustained as it diverts into the loops, where green color is observed in the figures.

#### 3.4.2. Velocity Contours of 60-Degree Tesla Valve (Case 6)

Figure 11a–c represent the velocity contours in the forward direction for the 60-degree Tesla valve, and the fluid flows from left to right (fetal bladder side to amniotic cavity side). The pressures applied to the fetal bladder side are 20, 30, 40 cm-H_2_O, and the pressures applied to the amniotic cavity side are 7, 12, 13 cm-H_2_O, respectively. As an example, Figure 11a represents the velocity contour of the valve with 20 cm-H_2_O pressure applied to the fetal bladder side and 7 cm-H_2_O pressure applied to the amniotic cavity side. In Figure 11a–c, the red color represents the highest velocity magnitude, the blue color represents the lowest velocity magnitude, and the colors orange, yellow, and green represent the velocity magnitude in between the highest and lowest. The highest velocity is at the fetal bladder side, due to the fluid flow is initiated from the fetal bladder. While the fluid enters from the fetal bladder to the tesla valve, the velocity decreases progressively to the amniotic cavity side in the straight channel. Within the loops, the velocity magnitude is very low, approaching zero, at the channel walls. It is a direct consequence of the no-slip boundary condition applied in the simulation, which assumes that the fluid velocity at the stationary solid boundary is zero.

Figure 11d–f illustrate the velocity contours in the backward direction for the 60-degree valve. The pressures applied to the amniotic cavity side are 20, 30, 40 cm-H_2_O, and the pressures on the fetal bladder side are 7, 12, 13 cm-H_2_O, respectively. As an example, Figure 11d illustrates the velocity contour of the valve with 7 cm-H_2_O pressure applied to the fetal bladder side and 20 cm-H_2_O pressure on the amniotic cavity side. The fluid is simulated to enter from the right side (amniotic cavity) to the left (the fetal bladder). As shown in Figure 11d–f, the red color is present near the entrance on the right side, indicating a high-velocity region. As the flow continues, it does not move in a straight channel; instead, it flows in a broken fashion, as seen in the figure of broken red zones. The red zones are not continuous, nor in a straight fashion, with some red appearing in the curved sections (loops). The fluid flow not only enters the loops, but also is not steady in the loops, as evidenced by the non-continuous red-orange areas. Therefore, the flow path is not sustained as it diverts into the loops, where green color is observed in the figures.

### 3.5. Interpretation of Pressure and Velocity Contours

The pressure contour shows the pressure gradient, constantly decreasing from the fetal bladder side to the amniotic cavity side in the forward direction. The pressure gradient is the driving force for the fluid to flow in the forward direction. Generally, the liquid flows from the high-pressure region to the low-pressure region.

The velocity contour is visualized as the result of the driving force driven by the pressure gradient and fluid movement. The velocity contour shows that, at any given cross-section, the velocity is typically highest in the center of the channel and decreases towards the walls due to viscous effects. This flow pattern throughout the channel, which is persistent by the pressure drop, represents the overall flow rate from the high-pressure fetal bladder to the low-pressure amniotic cavity.

The flow rate and velocity are lower for the 60-degree Tesla valve compared to the 50-degree Tesla valve, because the number of loops for the 60-degree valve is greater than that of the 50-degree valve. As the number of loops increases, the applied pressure difference should be higher to overcome the friction in the loops.

### 3.6. Velocity Vector Analysis

Based on the flow rate and diodicity analysis in the previous two sections, Tesla valve designs in cases 5 and 6 demonstrate the best performance regarding flow rate and diodicity. Therefore, this section will concentrate solely on velocity vector analysis for cases 5 and 6.

#### 3.6.1. Velocity Vectors for 50-Degree Tesla Valve (Case 5)

Figure 12a–c display the velocity vectors in the forward direction for the 50-degree Tesla valve, with fluid flowing from left to right (fetal bladder side to amniotic cavity side). The pressures on the fetal bladder side are 20, 30, and 40 cm-H_2_O, while the pressures on the amniotic cavity side are 7, 12, and 13 cm-H_2_O, respectively. For instance, Figure 12a illustrates the velocity vector of the 50-degree Tesla valve with 20 cm-H_2_O applied to the fetal bladder side and 7 cm-H_2_O to the amniotic cavity side. The flow moves from left to right, from the fetal bladder to the amniotic cavity. The color map in Figure 12a–c show that at the entrance of the Tesla valve’s straight channel, there is orange, indicating lower velocity, which transitions to red, indicating higher velocity. The peak velocities are 0.724 m-s, 0.881 m-s, and 1.13 m-s for the pressure gradients, respectively. of 20 to 7 cm-H_2_O, 30 to 12 cm-H_2_O, and 40 to 13 cm-H_2_O.

In the backward direction, flow starts from the amniotic cavity side, going toward the fetal bladder side due to higher pressure in the amniotic cavity. The pressures on the amniotic cavity side are 20, 30, and 40 cm-H_2_O, while on the fetal bladder side, they are 7, 12, and 13 cm-H_2_O, respectively. As shown in Figure 12d–f, the flow is hindered in this backward direction. Brief red jets appear at the loop entrances, and the main flow consists of large, energy-dissipating vortices shown by slower green and blue vectors, leading to lower peak velocities of about 0.67 m-s, 0.824 m-s, and 1.10 m-s for the pressure gradients of 20 to 7 cm-H_2_O, 30 to 12 cm-H_2_O, and 40 to 13 cm-H_2_O, respectively.

As a comparison, when applying a 13 cm-H_2_O pressure gradient to the 50-degree Tesla valve, the peak velocity magnitude in the forward direction is 0.724 m-s, compared to 0.67 m-s in the backward direction. The flow structure or behavior in the forward direction appears as a smooth red path versus a chaotic green and blue blending path in the backward direction. These differences in both velocity magnitude and flow structure or behavior for forward and backward directions quantitatively demonstrate and visually confirm the characteristic asymmetric flow resistance of the Tesla valve, which provides higher resistance to flow in the backward direction than in the forward direction, confirming its intended design used for the shunt.

#### 3.6.2. Velocity Vectors for 60-Degree Tesla Valve (Case 6)

Figure 13a–c display the velocity vectors in the forward direction for the 60-degree Tesla valve, with fluid flows from left to right (fetal bladder side to amniotic cavity side). The pressures on the fetal bladder side are 20, 30, 40 cm-H_2_O, and on the amniotic cavity side are 7, 12, 13 cm-H_2_O, respectively. For example, Figure 13a illustrates the velocity vector of the 60-degree Tesla valve with 20 cm-H_2_O applied to the fetal bladder side and 7 cm-H_2_O to the amniotic cavity side. The flow starts from left to right in the forward direction. The color maps in Figure 13a–c indicate that at the entrance of the straight channel, the color is orange, representing lower velocity, which then transitions to red, indicating higher velocity. The peak velocities are 0.693 m-s, 0.828 m-s, and 1.04 m-s for the pressure gradients of 20 to 7 cm-H_2_O, 30 to 12 cm-v, and 40 to 13 cm-H_2_O, respectively.

In the backward direction, flow is initiated from the right to the left side (amniotic cavity to fetal bladder) because the pressure on the amniotic cavity side is higher. The pressures on the amniotic cavity side are 20, 30, 40 cm-H_2_O, while on the fetal bladder side, they are 7, 12, 13 cm-H_2_O, respectively. As shown in Figure 13d–f, the flow is hindered in the backward direction. Brief red jets form at the loop entrances, and the dominant flow includes large, energy-dissipating vortices visualized by slower green and blue vectors, resulting in a lower peak velocity of approximately 0.688 m-s, 0.836 m-s, and 1.07 m-s for the three pressure gradients of 20 to 7 cm-H_2_O, 30 to 12 cm-H_2_O, and 40 to 13 cm-H_2_O, respectively.

As a comparison, when applying a 13 cm-H_2_O pressure gradient to the 60-degree Tesla valve, the peak velocity magnitude in the forward direction is 0.693 m-s, compared to 0.688 m-s in the backward direction. The flow structure or behavior in the forward direction appears as a smooth red path, while in the backward direction, it shows a chaotic green and blue blending path. These differences in both velocity magnitude and flow structure-behavior between the forward and backward directions quantitatively demonstrate and visually illustrate the characteristic asymmetric flow resistance of the Tesla valve, which provides higher resistance to flow in the backward direction than in the forward direction.

#### 3.6.3. Velocity Vectors Comparison Between 50-Degree and 60-Degree Tesla Valves

However, there is a difference in velocity magnitudes between 50-degree and 60-degree valves. For example, applying a 13 cm-H_2_O pressure gradient results in a maximum velocity of 0.724 m-s for the 50-degree valve in the forward direction, compared to 0.693 m-s for the 60-degree valve in the same direction. Since the 60-degree valve has more loops (6 loops) than the 50-degree valve (5 loops), this increased number of loops raises resistance to flow in both directions. The 50-degree valve produces a higher velocity than the 60-degree valve. The angle can influence fluid behavior, such as flow separation and vortex formation, which in turn affects flow speed.

## 4. Discussion

The significant finding in this study is how geometric extensions (0.5 mm, 1 mm, 2 mm) and angles (50 degrees and 60 degrees) in the Tesla valve design affect flow rate and diodicity. Simulations consistently show that certain configurations perform exceptionally well in achieving high forward flow rates and diodicity. This success is due to minimized flow disturbances and lower energy losses at the side branch junctions (loops) in the forward flow direction, which reduces overall flow resistance. Conversely, the different flow patterns seen in the backward direction—caused by high-pressure buildup and more prominent recirculation within the Tesla valve’s loops—highlight its natural ability to create asymmetric flow resistance, an essential trait for a passive flow regulator.

It is also important to note that the diodicity for this design, defined as the ratio of forward to reverse flow rate (Qf-Qr), is presented differently than in designs where diodicity is based on pressure drop ratios. However, this redefinition is more accurate for evaluating the valve’s efficiency in preventing backward flow under the natural pressure differences found in this physiological application. Its practical effectiveness in facilitating forward flow while hindering backward flow remains strong, which is crucial for the intended clinical application. In this context, maintaining a controlled pressure gradient between the fetal bladder and the amniotic cavity is essential to prevent high bladder resting tone and promote proper fetal bladder musculature development in conditions like LUTO. Additionally, while the simulations used the same pressure difference for both forward and backward flow conditions, real-time in vivo conditions for LUTO patients may show varying pressure differentials. Under such circumstances, this Tesla valve design could potentially be even more effective at preventing undesired backward flow.

### 4.1. Limitations and Validation

This study is entirely simulation-based, and the results require physical validation through experiments. Currently, several Tesla valves used as shunt inner channels have been 3D printed and are ready for fluid analysis experiments. Additionally, the fabrication of external meshing to form the shape with double mushroom geometries has begun. Once the Tesla valve is verified and the meshing is fabricated, we will combine both to create the complete shunt. External experiments will then be conducted to test shunt deployment and implantation in the lab, along with establishing safety procedures. Such studies will be published in a separate paper.

The current model does not justify complex biological variables such as dynamic intrauterine pressure changes, fetal movement, or varying bladder compliance, which could affect real-world performance. However, our successful simulation results will guide the next phase of development: rigorous in vitro testing to verify flow characteristics in a benchtop system, followed by essential in vivo testing in an animal model to confirm biocompatibility, safety, and therapeutic efficacy prior to clinical consideration.

### 4.2. Clinical Advantages over Existing Shunts

This passive flow regulation provides a key clinical advantage over existing devices. The Harrison shunt, which lacks a valve, risks pressure buildup and the need for re-intervention. In contrast, shunts like the Vortex use sensor-mediated valves, meaning their mechanism is not naturally driven by bladder pressure. Our Tesla valve-based vesico-amniotic shunt responds solely to pressure differences, which is expected to promote natural habituation and may support the development of fetal bladder musculature for voiding after birth.

## 5. Conclusions

In this study, we designed an optimal Tesla valve for fetal therapy in cases of LUTO, focusing on variations in the geometrical extensions of the fetal bladder side (0.5 mm, 1 mm, and 2 mm) and valve angles (50° and 60°), using CFD simulations. These simulations were run under boundary conditions that mimic physiological fetal fluid dynamics. Specifically, fetal thoracic pressures are usually lower than normal amniotic fluid pressure (28.55 cm-H_2_O), but can rise significantly in the presence of large effusions, reaching up to 53 cm-H_2_O. Among the tested configurations, the design with a 0.5 mm extension on the fetal side and a 2 mm extension on the amniotic cavity side showed the highest flow rate and diodicity, indicating better performance in promoting unidirectional flow. Detailed analysis of pressure contours, velocity fields, and vector plots revealed that the 50° valve angle outperformed the 60° angle configuration. This improvement is due to less formation of flow loops at 50°, which reduces resistance and increases flow velocity under the same physiological conditions.

Overall, the Tesla valve configuration with a 0.5 mm fetal side extension, a 2 mm amniotic cavity side extension, and a 50° valve angle was identified as the most effective design for fetal shunting in LUTO. This design showed optimal flow characteristics and diodicity under clinically relevant conditions, supporting its potential use in fetal therapy.

## Figures and Tables

**Figure 1 bioengineering-12-01126-f001:**
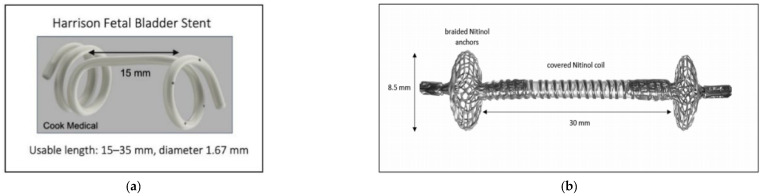
(**a**) Harrison Fetal Bladder Stent [12]. (**b**) Vortex Shunt [15].

**Figure 2 bioengineering-12-01126-f002:**
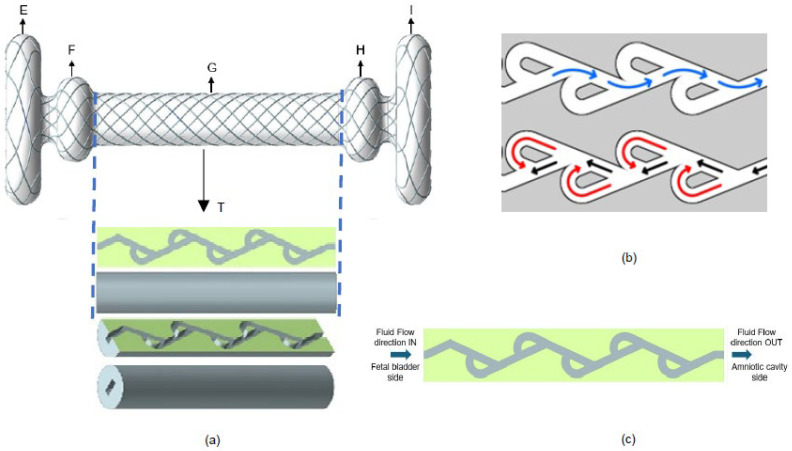
(**a**) New fetal shunt design with Tesla valve inside. (**b**) Flow directions of Tesla valve in forward flow (blue arrows) and backward flow (black and red arrows). (**c**) Sectional view of Tesla valve body.

**Figure 3 bioengineering-12-01126-f003:**
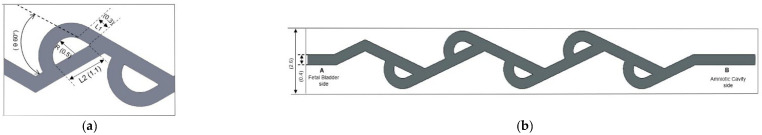
(**a**) Single Tesla valve loop 2D geometry with major geometric parameters (dimensions in millimeters). (**b**) Full-length Tesla valve illustration.

**Figure 4 bioengineering-12-01126-f004:**
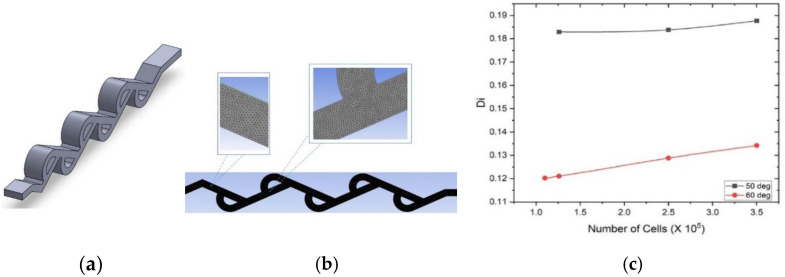
(**a**) 3D model of 50-degree Tesla valve. (**b**) Unstructured triangular mesh of 50-degree Tesla valve. (**c**) Di vs. numbers of cells of 50- and 60-degree Tesla valves.

**Figure 5 bioengineering-12-01126-f005:**
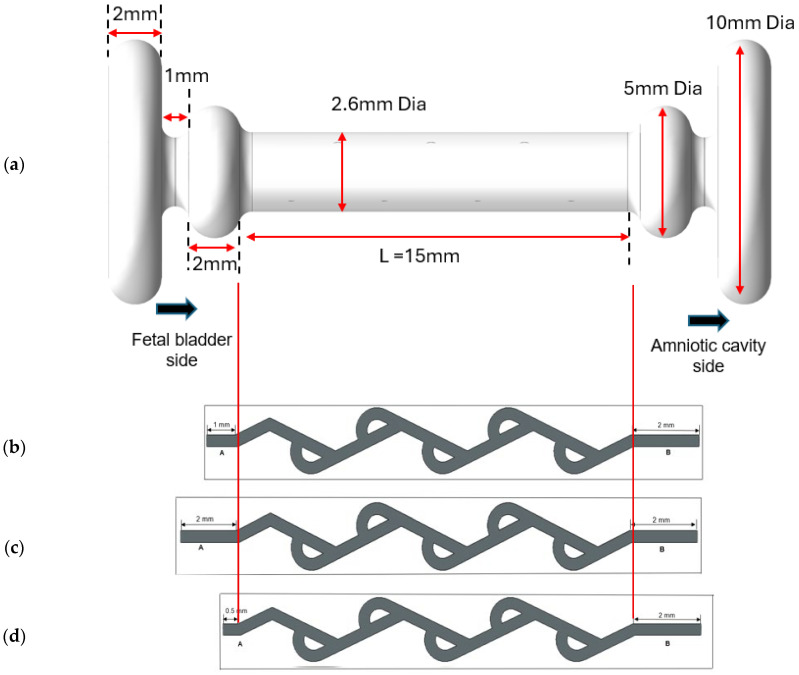
(**a**) Model of double mushroom shunt design (meshing is not shown). (**b**) Case 1 with a 1 mm extension on the fetal bladder side and a 2 mm extension on the amniotic cavity side of a 50-degree Tesla valve. (**c**) Case 3 with a 2 mm extension on both sides of a 50-degree Tesla valve. (**d**) Case 3 with a 0.5 mm on the fetal bladder side and a 2 mm extension on the amniotic cavity side of a 50-degree Tesla valve.

**Figure 6 bioengineering-12-01126-f006:**
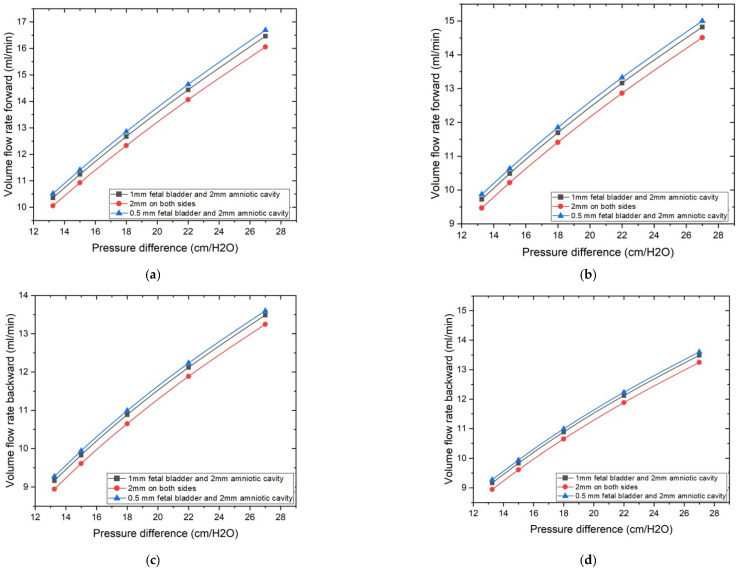
Geometric extension dimensions on the impact of volume flow rate vs. pressure differences. (**a**) 50-degree valve: Volume flow rate vs. pressure difference in forward direction. (**b**) 60-degree valve: Volume flow rate vs. pressure difference in forward direction. (**c**) 50-degree valve: Volume flow rate vs. pressure difference in backward direction. (**d**) 60-degree valve: Volume flow rate vs. pressure difference in backward direction.

**Figure 7 bioengineering-12-01126-f007:**
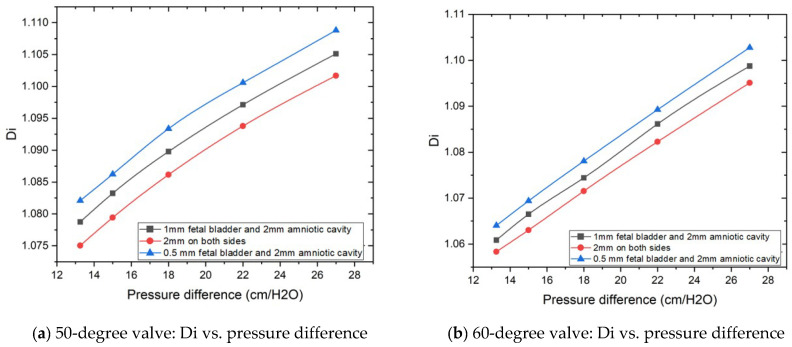
Geometric extension dimensions on the impact of diodicity based on pressure differences.

**Figure 8 bioengineering-12-01126-f008:**
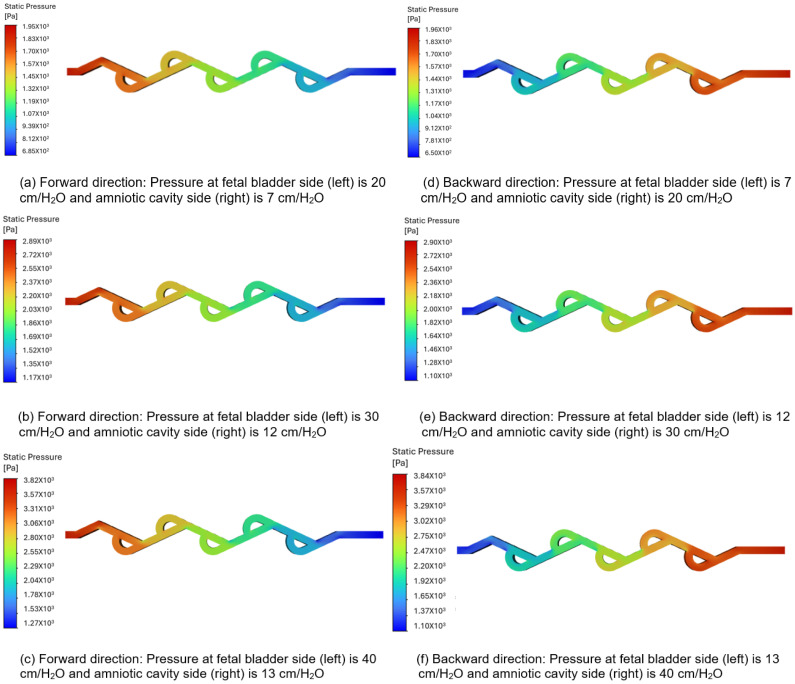
Static pressure contours of a 50-degree valve in forward and backward directions.

**Figure 9 bioengineering-12-01126-f009:**
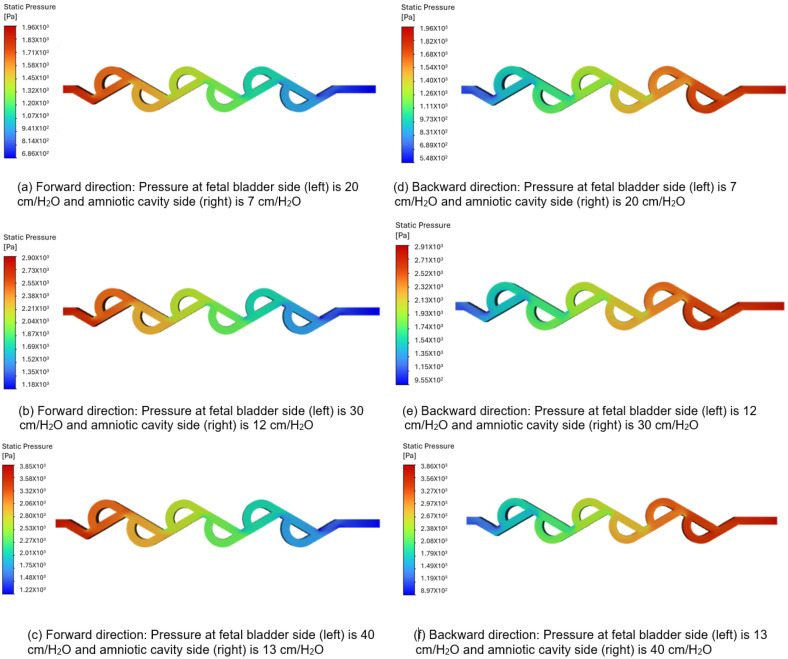
Static pressure contours of a 60-degree valve in forward and backward directions.

**Figure 10 bioengineering-12-01126-f010:**
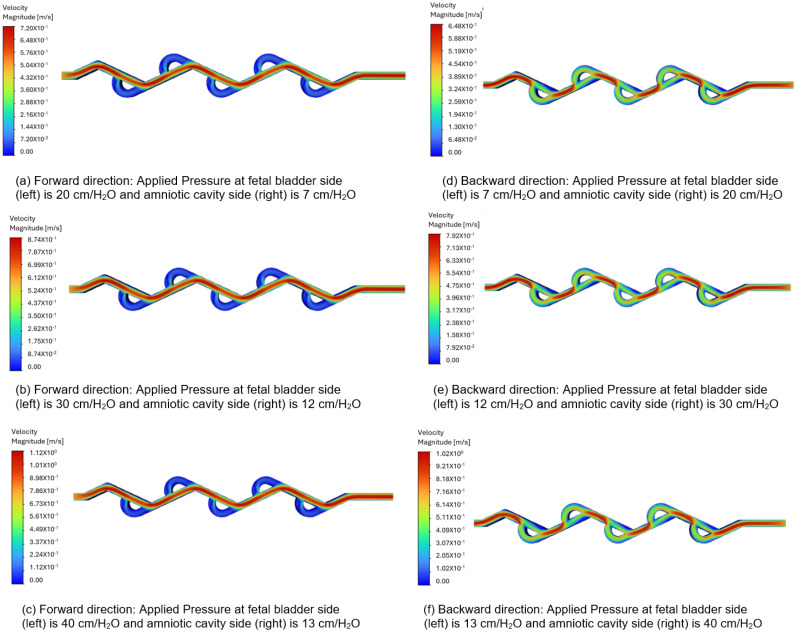
Comparison of static velocity contours of 50-degree Tesla valve with geometric design case 5 and applied three pressure differences.

**Figure 11 bioengineering-12-01126-f011:**
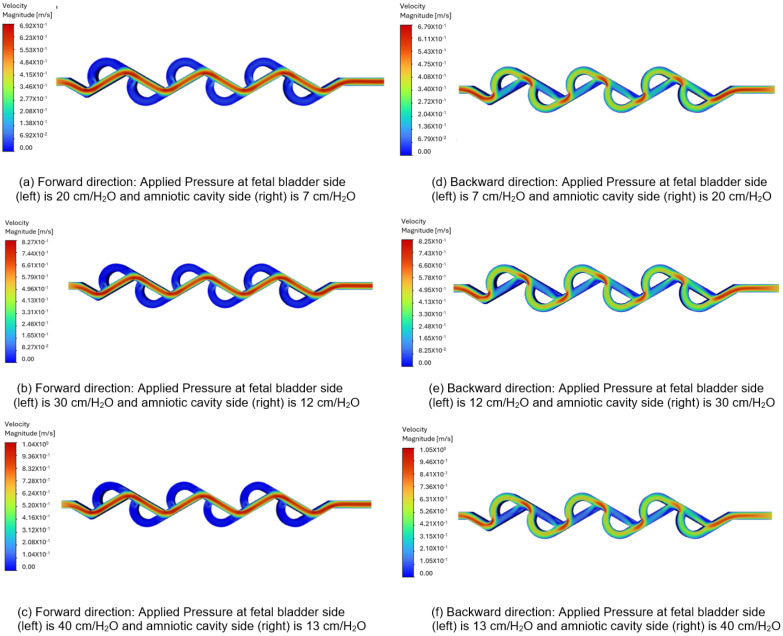
Comparison of static velocity contours of 60-degree Tesla valve with geometric design case 6 and applied three pressure differences.

**Figure 12 bioengineering-12-01126-f012:**
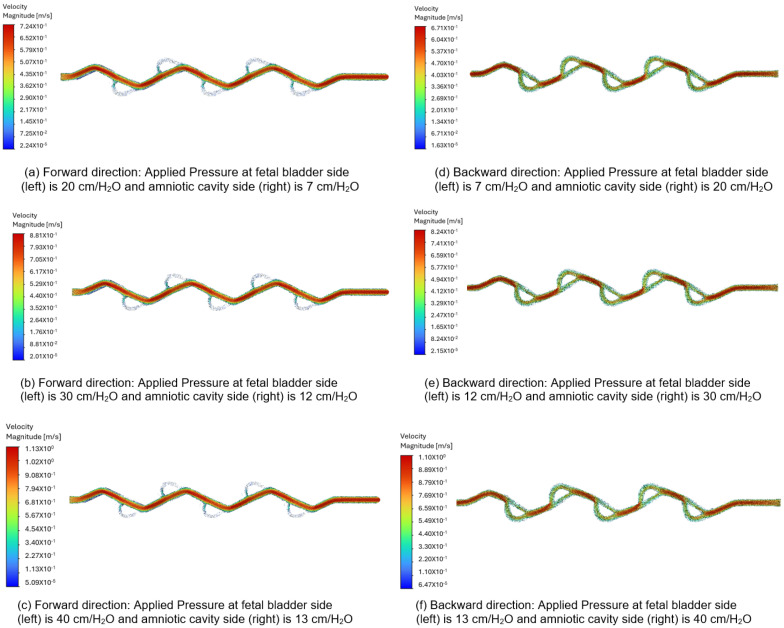
Comparison of the velocity vector for a 50-degree Tesla valve with geometric design case 5 and applied three pressure differences.

**Figure 13 bioengineering-12-01126-f013:**
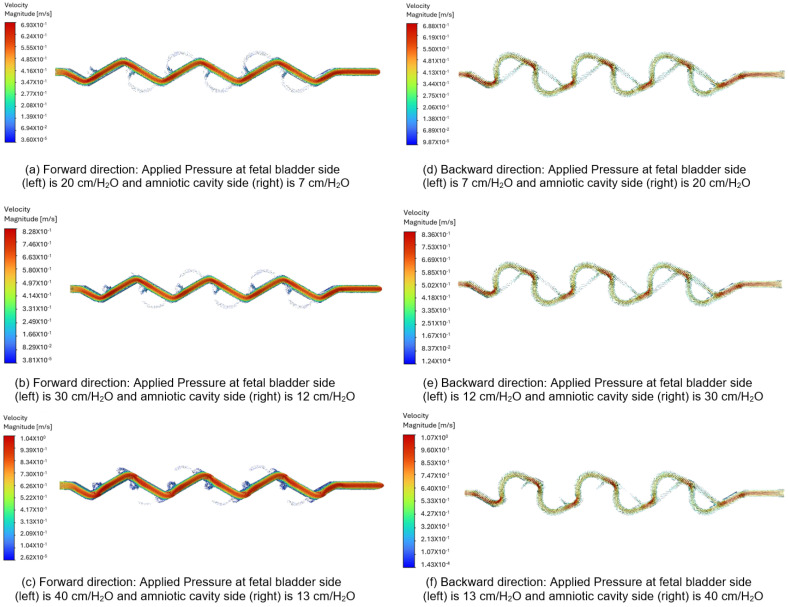
Comparison of the velocity vector for a 60-degree Tesla valve with geometric design case 6 and applied three pressure differences.

## Data Availability

The original contributions presented in this study are included in the article. Further inquiries can be directed to the corresponding author.

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
