# Peer review of "Computational Study of Tesla Valve Design for Vesico-Amniotic Shunt to Manage Lower Urinary Tract Obstruction and Pleural Effusion"

_bioengineering, 2025, doi:10.3390/bioengineering12101126_

Round 1

Reviewer 1 Report

Comments and Suggestions for Authors

Compliments to the authors for presenting an interesting Tesla Valve design used for Fetal Shunt.

Comments:

  1. The title needs change. Fetal shunt is a broad term used not just for vesico-amniotic shunt. So to say fetal shunt would improve the bladder musculature development in LUTO may not be totally correct. The fact that the shunt would improve bladder musculature development is also an assumption and it has not been proven (except few animal studies (https://doi.org/10.1016/j.jpedsurg.2008.08.058). Vesico-amniotic shunts usually are placed with an intent to improve post-natal renal function and not for bladder musculature development (Ref: Kohl, Thomas et al. Journal of Pediatric Urology, Volume 18, Issue 2, 116 – 126)
  2. You write that (line 35) pleural effusion compresses the fetal head and resultant fetal cardiac failure; I think this is a typo error.
  3. Line 36: do you want to call it feto-amniotic shunt or vesicoamniotic shunt?
  4. Line 94: you mention, amniotic cavity on the maternal side; I think just saying shunt extends into the amniotic cavity is enough.

Author Response

see attached file for our response to your comments. Thank you. 

Reviewer 2 Report

Comments and Suggestions for Authors

The manuscript introduces an innovative simulation-based study that applies Tesla valve principles to the design of a fetal shunt for lower urinary tract obstruction (LUTO). The topic is clinically relevant and technically engaging, and the manuscript is generally well-structured. However, several critical issues need to be addressed before the paper can be considered for publication. My comments and suggestions for improvement are as follows:

  1. Title – Please avoid using abbreviations in the title. Please spell out LUTO in full.
  2. Introduction – The introduction should more precisely describe the underlying diseases that cause fetal lower urinary tract obstruction (e.g., posterior urethral valves, urethral atresia, cloacal malformations) and pleural effusions (e.g., congenital chylothorax, cardiac anomalies, chromosomal abnormalities). This would provide essential clinical context for readers. Please add a statement and appropriate references (e.g., for LUTO – doi: 10.3390/molecules29143294 and DOI: 10.1002/uog.21994 for pleural effusions).
  3. Methods – Although the paper provides technical details on CFD modeling, the description is disorganized. It is unclear which solver or software was used, how the boundary conditions were set, and what assumptions were made (e.g., Newtonian versus non-Newtonian fluids, steady versus pulsatile flow). The methods section should be better organized and more transparent to enhance reproducibility and make it more accessible to a wider biomedical audience.
  4. Validation – The study is entirely simulation-based. This should be clearly stated as a key limitation, acknowledging that validation through in-vitro or animal models will be necessary before clinical application. Currently, the lack of validation limits the strength of the conclusions.
  5. Limitations of the study – The discussion briefly addresses the design implications but does not sufficiently expand on the limitations of the simulation approach. The authors should clearly state that biological variability (e.g., dynamic intrauterine pressure changes, fetal movement, bladder compliance) could affect the performance of the proposed design. Additionally, the authors did not clearly emphasize the absence of external validation. Since this is a purely simulation-based study, the lack of in vitro or in vivo testing should be explicitly discussed as a major limitation. Highlighting this will set appropriate expectations and outline the necessary next steps before clinical translation.
  6. Clinical implications – The authors should include a more detailed discussion on the feasibility of manufacturing, implantation, and safety of the proposed shunt, and compare their design more explicitly with existing devices (Harrison stent, Vortex shunt) in terms of expected benefits.

Author Response

see attached file for our responses to your comments. thank you. 

Round 2

Reviewer 2 Report

Comments and Suggestions for Authors

The authors adequately addressed my objections and improved the paper. I believe the manuscript can be accepted in its current form.